# Searching for Efficient Multi-Scale Architectures for Dense Image Prediction

**Liang-Chieh Chen**    **Maxwell D. Collins**    **Yukun Zhu**    **George Papandreou**
**Barret Zoph**    **Florian Schroff**    **Hartwig Adam**    **Jonathon Shlens**
`Google Inc.`

## Abstract

The design of neural network architectures is an important component for achieving state-of-the-art performance with machine learning systems across a broad array of tasks. Much work has endeavored to design and build architectures automatically through clever construction of a search space paired with simple learning algorithms. Recent progress has demonstrated that such meta-learning methods may exceed scalable human-invented architectures on image classification tasks. An open question is the degree to which such methods may generalize to new domains. In this work we explore the construction of meta-learning techniques for dense image prediction focused on the tasks of scene parsing, person-part segmentation, and semantic image segmentation. Constructing viable search spaces in this domain is challenging because of the multi-scale representation of visual information and the necessity to operate on high resolution imagery. Based on a survey of techniques in dense image prediction, we construct a recursive search space and demonstrate that even with efficient random search, we can identify architectures that outperform human-invented architectures and achieve state-of-the-art performance on three dense prediction tasks including 82.7% on Cityscapes (street scene parsing), 71.3% on PASCAL-Person-Part (person-part segmentation), and 87.9% on PASCAL VOC 2012 (semantic image segmentation). Additionally, the resulting architecture is more computationally efficient, requiring half the parameters and half the computational cost as previous state of the art systems.

## 1  Introduction

The resurgence of neural networks in machine learning has shifted the emphasis for building state-of-the-art systems in such tasks as image recognition [44, 84, 83, 34], speech recognition [36, 8], and machine translation [88, 82] towards the design of neural network architectures. Recent work has demonstrated successes in automatically designing network architectures, largely focused on single-label image classification tasks [100, 101, 52] (but see [100, 65] for language tasks). Importantly, in just the last year such meta-learning techniques have identified architectures that exceed the performance of human-invented architectures for large-scale image classification problems [101, 52, 68].

Image classification has provided a great starting point because much research effort has identified successful network motifs and operators that may be employed to construct search spaces for architectures [52, 68, 101]. Additionally, image classification is inherently multi-resolution whereby fully convolutional architectures [77, 58] may be trained on low resolution images (with minimal computational demand) and be transferred to high resolution images [101].

Although these results suggest opportunity, the real promise depends on the degree to which meta-learning may extend into domains beyond image classification. In particular, in the image domain, many important tasks such as semantic image segmentation [58, 11, 97], object detection [71, 21],

and instance segmentation [20, 33, 9] rely on high resolution image inputs and multi-scale image representations. Naïvely porting ideas from image classification would not suffice because (1) the space of network motifs and operators differ notably from systems that perform classification and (2) architecture search must inherently operate on high resolution imagery. This final point makes previous approaches computationally intractable where transfer learning from low to high image resolutions was critical [101].

In this work, we present the first effort towards applying meta-learning to dense image prediction (Fig. 1) – largely focused on the heavily-studied problem of scene labeling. Scene labeling refers to the problem of assigning semantic labels such as "person" or "bicycle" to every pixel in an image. State-of-the-art systems in scene labeling are elaborations of convolutional neural networks (CNNs) largely structured as an encoder-decoder in which various forms of pooling, spatial pyramid structures [97] and atrous convolutions [11] have been explored. The goal of these operations is to build a multi-scale representation of a high resolution image to densely predict pixel values (*e.g.*, stuff label, object label, *etc*.). We leverage off this literature in order to construct a search space over network motifs for dense prediction. Additionally, we perform an array of experiments to demonstrate how to construct a computationally tractable and simple proxy task that may provide predictive information on multi-scale architectures for high resolution imagery.

We find that an effective random search policy provides a strong baseline [5, 30] and identify several candidate network architectures for scene labeling. In experiments on the Cityscapes dataset [18], we find architectures that achieve 82.7% mIOU accuracy, exceeding the performance of human-invented architectures by 0.7% [6]. For reference, note that achieving gains on the Cityscapes dataset is challenging as the previous academic competition elicited gains of 0.8% in mIOU from [97] to [6] over more than one year. Additionally, this same network applied to other dense prediction tasks such as person-part segmentation [16] and semantic image segmentation [24] surpasses state-of-the-art results [25, 93] by 3.7% and 1.7% in absolute percentage, respectively (and comparable to concurrent works [14, 96, 48] on VOC 2012). This is the first time to our knowledge that a meta-learning algorithm has matched state-of-the-art performance using architecture search techniques on dense image prediction problems. Notably, the identified architecture operates with half the number of trainable parameters and roughly half the computational demand (in Multiply-Adds) as previous state-of-the-art systems [14], when employing the powerful Xception [17, 67, 14] as network backbone[1].

## 2 Related Work

### 2.1 Architecture search

Our work is motivated by the neural architecture search (NAS) method [100, 101], which trains a controller network to generate neural architectures. In particular, [101] transfers architectures learned on a proxy dataset [43] to more challenging datasets [73] and demonstrates superior performance over many human-invented architectures. Many parallel efforts have employed reinforcement learning [3, 99], evolutionary algorithms [81, 69, 59, 90, 53, 68] and sequential model-based optimization [61, 52] to learn network structures. Additionally, other works focus on successively increasing model size [7, 15], sharing model weights to accelerate model search [65], or a continuous relaxation of the architecture representation [54]. Note that our work is complimentary and may leverage all of these advances in search techniques to accelerate the search and decrease computational demand.

Critically, all approaches are predicated on constructing powerful but tractable architecture search spaces. Indeed, [52, 101, 68] find that sophisticated learning algorithms achieve superior results; however, even random search may achieve strong results if the search space is not overly expansive. Motivated by this last point, we focus our efforts on developing a tractable and powerful search space for dense image prediction paired with efficient random search [5, 30].

Recently, [75, 27] proposed methods for embedding an exponentially large number of architectures in a grid arrangement for semantic segmentation tasks. In this work, we instead propose a novel recursive search space and simple yet predictive proxy tasks aimed at finding effective architectures for dense image prediction.

## 2.2 Multi-scale representation for dense image prediction

State-of-the-art solutions for dense image predictions derive largely from convolutional neural networks [46]. A critical element of building such systems is supplying global features and context information to perform pixel-level classification [35, 78, 41, 45, 31, 92, 60, 19, 63]. Several approaches exist for how to efficiently encode the multi-scale context information in a network architecture: (1) designing models that take as input an image pyramid so that large scale objects are captured by the downsampled image [26, 66, 23, 50, 13, 11], (2) designing models that contain encoder-decoder structures [2, 72, 49, 28, 64, 93, 96], or (3) designing models that employ a multi-scale context module, *e.g.*, DenseCRF module [42, 4, 10, 98, 50, 76], global context [56, 95], or atrous convolutions deployed in cascade [57, 94, 12] or in parallel [11, 12]. In particular, PSPNet [97] and DeepLab [12, 14] perform spatial pyramid pooling at several hand-designed grid scales.

A common theme in the dense prediction literature is how to best tune an architecture to extract context information. Several works have focused on sampling rates in atrous convolution to encode multi-scale context [37, 29, 77, 62, 10, 94, 11]. DeepLab-v1 [10] is the first model that enlarges the sampling rate to capture long range information for segmentation. The authors of [94] build a context module by gradually increasing the rate on top of belief maps, the final CNN feature maps that contain output channels equal to the number of predicted classes. The work in [87] employs a hybrid of rates within the last two blocks of ResNet [34], while Deformable ConvNets [22] proposes the *deformable convolution* which generalizes atrous convolution by learning the rates. DeepLab-v2 [11] and DeepLab-v3 [12] employ a module, called ASPP (atrous spatial pyramid pooling module), which consists of several *parallel* atrous convolutions with different rates, aiming to capture different scale information. Dense-ASPP [91] proposes to build the ASPP module in a densely connected manner. We discuss below how to construct a search space that captures all of these features.

## 3 Methods

Two key components for building a successful architecture search method are the design of the search space and the design of the proxy task [100, 101]. Most of the human expertise shifts from architecture design to the construction of a search space that is both expressive and tractable. Likewise, identifying a proxy task that is both predictive of the large-scale task but is extremely quick to run is critical for searching this space efficiently.

### 3.1 Architecture search space

The goal of architecture search space is to design a space that may express a wide range of architectures, but also be tractable enough for identifying good models. We start with the premise of building a search space that may express all of the state-of-the-art dense prediction and segmentation models previously discussed (e.g. [12, 97] and see Sec. 2 for more details).

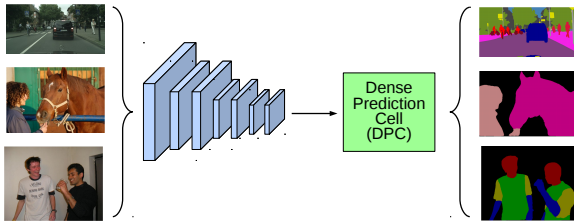

We build a recursive search space to encode multi-scale context information for dense prediction tasks that we term a Dense Prediction Cell (DPC). The cell is represented by a directed acyclic graph (DAG) which consists of $\mathcal{B}$ branches and each branch maps one input tensor to another output tensor. In preliminary experiments we found that $\mathcal{B} = 5$ provides a good trade-off between flexibility and computational tractability (see Sec. 5 for more discussion).

Figure 1: Schematic diagram of architecture search for dense image prediction. Example tasks explored in this paper include scene parsing [18], semantic image segmentation [24] and person-part segmentation [16].

We specify a branch $b_i$ in a DPC as a 3-tuple, $(X_i, OP_i, Y_i)$, where $X_i \in \mathcal{X}_i$ specifies the input tensor, $OP_i \in \mathcal{OP}$ specifies the operation to apply to input $X_i$, and $Y_i$ denotes the output tensor. The final output, $Y$, of the DPC is the concatenation of all branch outputs, *i.e.*, $Y = concat(Y_1, Y_2, \ldots, Y_{\mathcal{B}})$, allowing us to exploit all the learned information from each branch. For branch $b_i$, the set of possible inputs, $\mathcal{X}_i$, is equal to the last network backbone feature maps, $\mathcal{F}$, plus all outputs obtained by

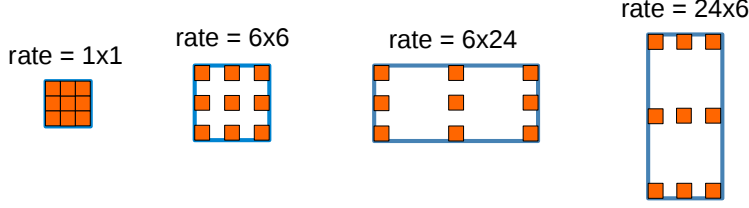

Figure 2: Diagram of the search space for atrous convolutions. $3 \times 3$ atrous convolutions with sampling rates $r_h \times r_w$ to capture contexts with different aspect ratios. From left to right: standard convolution ($1 \times 1$), equal expansion ($6 \times 6$), short and fat ($6 \times 24$) and tall and skinny ($24 \times 6$).

previous branches, $Y_1, \ldots, Y_{i-1}$, *i.e.*, $\mathcal{X}_i = \{\mathcal{F}, Y_1, \ldots, Y_{i-1}\}$. Note that $\mathcal{X}_1 = \{\mathcal{F}\}$, *i.e.*, the first branch can only take $\mathcal{F}$ as input.

The operator space, $\mathcal{OP}$, is defined as the following set of functions:

- Convolution with a $1 \times 1$ kernel.
- $3 \times 3$ atrous separable convolution with rate $r_h \times r_w$, where $r_h$ and $r_w \in \{1, 3, 6, 9, \ldots, 21\}$.
- Average spatial pyramid pooling with grid size $g_h \times g_w$, where $g_h$ and $g_w \in \{1, 2, 4, 8\}$.

For the spatial pyramid pooling operation, we perform average pooling in each grid. After the average pooling, we apply another $1 \times 1$ convolution followed by bilinear upsampling to resize back to the same spatial resolution as input tensor. For example, when the pooling grid size $g_h \times g_w$ is equal to $1 \times 1$, we perform image-level average pooling followed by another $1 \times 1$ convolution, and then resize back (*i.e.*, tile) the features to have the same spatial resolution as input tensor.

We employ separable convolution [79, 85, 86, 17, 38] with 256 filters for all the convolutions, and decouple sampling rates in the $3 \times 3$ atrous separable convolution to be $r_h \times r_w$ which allows us to capture object scales with different aspect ratios. See Fig. 2 for an example.

The resulting search space may encode all leading architectures but is more diverse as each branch of the cell may build contextual information through parallel or cascaded representations. The potential diversity of the search space may be expressed in terms of the total number of potential architectures. For $i$-th branch, there are $i$ possible inputs, including the last feature maps produced by the network backbone (*i.e.*, $\mathcal{F}$) as well as all the previous branch outputs (*i.e.*, $Y_1, \ldots, Y_{i-1}$), and $1 + 8 \times 8 + 4 \times 4 = 81$ functions in the operator space, resulting in $i \times 81$ possible options. Therefore, for $\mathcal{B} = 5$ branches, the search space contains $\mathcal{B}! \times 81^{\mathcal{B}} \approx 4.2 \times 10^{11}$ configurations.

## 3.2 Architecture search

The model search framework builds on top of an efficient optimization service [30]. It may be thought of as a black-box optimization tool whose task is to optimize an objective function $f : \boldsymbol{b} \to \mathbb{R}$ with a limited evaluation budget, where in our case $\boldsymbol{b} = \{b_1, b_2, \ldots, b_{\mathcal{B}}\}$ is the architecture of DPC and $f(\boldsymbol{b})$ is the pixel-wise mean intersection-over-union (mIOU) [24] evaluated on the dense prediction dataset. The black-box optimization refers to the process of generating a sequence of $\boldsymbol{b}$ that approaches the global optimum (if any) as fast as possible. Our search space size is on the order of $10^{11}$ and we adopt the *random search* algorithm implemented by Vizier [30], which basically employs the strategy of sampling points $\boldsymbol{b}$ uniformly at random as well as sampling some points $\boldsymbol{b}$ near the currently best observed architectures. We refer the interested readers to [30] for more details. Note that the *random search* algorithm is a simple yet powerful method. As highlighted in [101], random search is competitive with reinforcement learning and other learning techniques [52].

## 3.3 Design of the proxy task

Naïvely applying architecture search to a dense prediction task requires an inordinate amount of computation and time, as the search space is large and training a candidate architecture is time-consuming. For example, if one fine-tunes the entire model with a single dense prediction cell (DPC) on the Cityscapes dataset, then training a candidate architecture with 90K iterations requires 1+ week

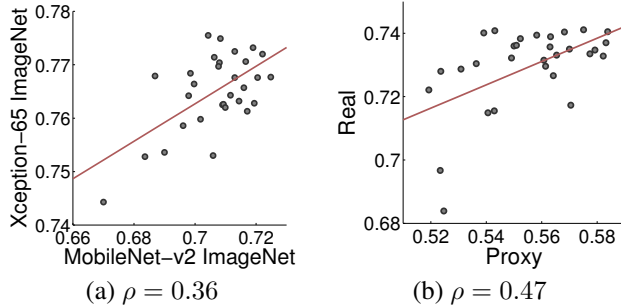

(a) $\rho = 0.36$        (b) $\rho = 0.47$

Figure 3: Measuring the fidelity of proxy tasks for a dense prediction cell (DPC) in a reduced search space. In preliminary search spaces, a comparison of (a) small to large network backbones, and (b) proxy versus large-scale training with MobileNet-v2 backbone. $\rho$ is Spearman's rank correlation coefficient.

with a single P100 GPU. Therefore, we focus on designing a proxy task that is (1) fast to compute and (2) may predict the performance in a large-scale training setting.

Image classification employs low resolution images [43] as a fast proxy task for high-resolution [73]. This proxy task does not work for dense image prediction where high resolution imagery is critical for conveying multi-scale context information. Therefore, we propose to design the proxy dataset by (1) employing a smaller network backbone and (2) caching the feature maps produced by the network backbone on the training set and directly building a single DPC on top of it. Note that the latter point is equivalent to not back-propagating gradients to the network backbone in the real setting In addition, we elect for *early stopping* by not training candidate architectures to convergence. In our experiments, we only train each candidate architecture with 30K iterations. In summary, these two design choices result in a proxy task that runs in 90 minutes on a GPU cutting down the computation time by $100+$-fold but is predictive of larger tasks ($\rho \geq 0.4$).

After performing architecture search, we run a reranking experiment to more precisely measure the efficacy of each architecture in the large-scale setting [100, 101, 68]. In the reranking experiments, the network backbone is fine-tuned and trained to full convergence. The new top architectures returned by this experiment are presented in this work as the best DPC architectures.

## 4 Results

We demonstrate the effectiveness of our proposed method on three dense prediction tasks that are well studied in the literature: scene parsing (Cityscapes [18]), person part segmentation (PASCAL-Person-Part [16]), and semantic image segmentation (PASCAL VOC 2012 [24]). Training and evaluation protocols follow [12, 14]. In brief, the network backbone is pre-trained on the COCO dataset [51]. The training protocol employs a polynomial learning rate [56] with an initial learning rate of $0.01$, large crop sizes (*e.g.*, $769 \times 769$ on Cityscapes and $513 \times 513$ on PASCAL images), fine-tuned batch normalization parameters [40] and small batch training (batch size = 8, 16 for proxy and real tasks, respectively). For evaluation and architecture search, we employ a single image scale. For the final results in which we compare against other state-of-the-art systems (Tab. 2, Tab. 3 and Tab. 4), we perform evaluation by averaging over multiple scalings of a given image.

### 4.1 Designing a proxy task for dense prediction

The goal of a proxy task is to identify a problem that is quick to evaluate but provides a predictive signal about the large-scale task. In the image classification work, the proxy task was classification on low resolution (e.g. $32 \times 32$) images [100, 101]. Dense prediction tasks innately require high resolution images as training data. Because the computational demand of convolutional operations scale as the number of pixels, another meaningful proxy task must be identified.

We approach the problem of proxy task design by focusing on speed and predictive ability. As discussed in Sec. 3, we employ several strategies for devising a fast and predictive proxy task to speed up the evaluation of a model from $1+$ week to 90 minutes on a single GPU. In these preliminary

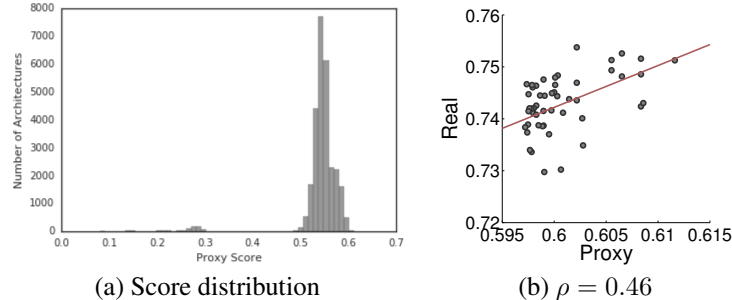

(a) Score distribution          (b) $\rho = 0.46$

Figure 4: Measuring the fidelity of the proxy tasks for a dense prediction cell (DPC) in the full search space. (a) Score distribution on the proxy task. The search algorithm is able to explore a diversity of architectures. (b) Correlation of the found top-50 architectures between the proxy dataset and large-scale training with MobileNet-v2 backbone. $\rho$ is Spearman's rank correlation coefficient.

experiments, we demonstrate that these strategies provide an instructive signal for predicting the efficacy of a given architecture.

To minimize stochastic variation due to sampling architectures, we first construct an extremely small search space containing only 31 architectures[2] in which we may exhaustively explore performance. We perform the experiments and subsequent architecture search on Cityscapes [18], which features large variations in object scale across 19 semantic labels.

Following previous state-of-the-art segmentation models, we employ the Xception architecture [17, 67, 14] for the large-scale setting. We first asked whether a smaller network backbone, MobileNet-v2 [74] provides a strong signal of the performance of the large network backbone (Fig. 3a). MobileNet-v2 consists of roughly $\frac{1}{20}$ the computational cost and cuts down the backbone feature channels from 2048 to 320 dimensions. We indeed find a rank correlation ($\rho = 0.36$) comparable to learned predictors [52], suggesting that this may provide a reasonable substitute for the proxy task. We next asked whether employing a fixed and cached set of activations correlates well with training end-to-end. Fig. 3b shows that a higher rank correlation between cached activations and training end-to-end for COCO pretrained MobileNet-v2 backbone ($\rho = 0.47$). The fact that these rank correlations are significantly above chance rate ($\rho = 0$) indicates that these design choices provide a useful signal for large-scale experiments (*i.e.*, more expensive network backbone) comparable to learned predictors [52, 101] (for reference, $\rho \in [0.41, 0.47]$ in the last stage of [52]) as well as a fast proxy task.

## 4.2 Architecture search for dense prediction cells

We deploy the resulting proxy task, with our proposed architecture search space, on Cityscapes to explore 28K DPC architectures across 370 GPUs over one week. We employ a simple and efficient random search [5, 30] and select the top 50 architectures (w.r.t. validation set performance) for re-ranking based on fine-tuning the entire model using MobileNet-v2 network backbone. Fig. 4a highlights the distribution of performance scores on the proxy dataset, showing that the architecture search algorithm is able to explore a diversity of architectures. Fig. 4b demonstrates the correlation of the found top-50 DPCs between the original proxy task and the re-ranked scores. Notably, the top model identified with re-ranking was the $12^{th}$ best model as measured by the proxy score.

Fig. 5a provides a schematic diagram of the top DPC architecture identified (see Fig. 6 for the next best performing ones). Following [39] we examine the $L1$ norm of the weights connecting each branch (via a $1 \times 1$ convolution) to the output of the top performing DPC in Fig. 5b. We observe that the branch with the $3 \times 3$ convolution (rate $= 1 \times 6$) contributes most, whereas the branches with large rates (*i.e.*, longer context) contribute less. In other words, information from image features in closer proximity (i.e. final spatial scale) contribute more to the final outputs of the network. In contrast, the worst-performing DPC (Fig. 6c) does not preserve fine spatial information as it cascades four branches after the global image pooling operation.

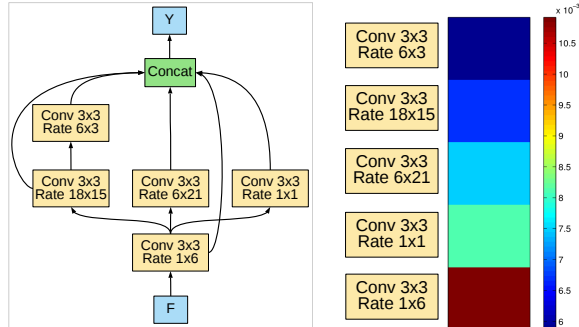

Figure 5: Schematic diagram of top ranked DPC (left) and average absolute filter weights ($L1$ norm) for each operation (right).

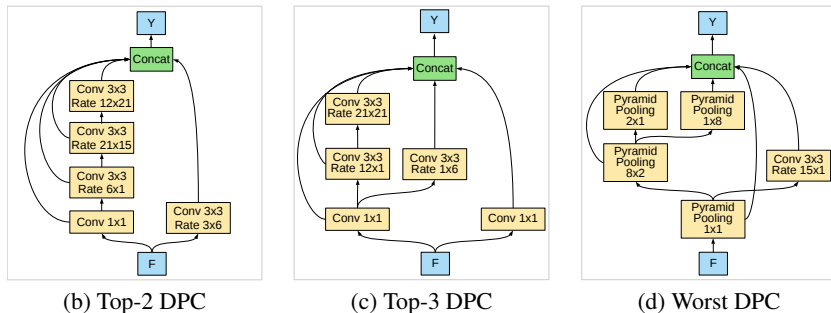

(b) Top-2 DPC          (c) Top-3 DPC          (d) Worst DPC

Figure 6: Diversity of DPCs explored in architecture search. (b-d) Top-2, Top-3 and worst DPCs.

| Network Backbone | Module | Params | MAdds | mIOU (%) |
|---|---|---|---|---|
| MobileNet-v2 | ASPP [12] | 0.25M | 2.82B | 73.97 |
| MobileNet-v2 | DPC | 0.36M | 3.00B | **75.38** |
| Modified Xception | ASPP [12] | 1.59M | 18.12B | 80.25 |
| Modified Xception | DPC | 0.81M | 6.84B | **80.85** |

Table 1: Cityscapes *validation* set performance (labeling IOU) across different network backbones (output stride = 16). ASPP is the previous state-of-the-art system [12] and DPC indicates this work. Params and MAdds indicate the number of parameters and number of multiply-add operations in each multi-scale context module.

### 4.3 Performance on scene parsing

We train the best learned DPC with MobileNet-v2 [74] and modified Xception [17, 67, 14] as network backbones on Cityscapes training set [18] and evaluate on the validation set. The network backbone is pretrained on the COCO dataset [51] for this and all subsequent experiments. Fig. 1 in the supplementary material shows qualitative results of the predictions from the resulting architecture. Quantitative results in Tab. 1 highlight that the learned DPC provides $1.4\%$ improvement on the validation set when using MobileNet-v2 network backbone and a $0.6\%$ improvement when using the larger modified Xception network backbone. Furthermore, the best DPC only requires half of the parameters and $38\%$ of the FLOPS of the previous state-of-the-art dense prediction network [14] when using Xception as network backbone. We note the computation saving results from the cascaded structure in our top-1 DPC, since the feature channels of Xception backbone is 2048 and thus it is expensive to directly build *parallel* operations on top of it (like ASPP).

We next evaluate the performance on the test set (Tab. 2). DPC sets a new state-of-the-art performance of $82.7\%$ mIOU – an $0.7\%$ improvement over the state-of-the-art model [6]. This model outperforms other state-of-the-art models across 11 of the 19 categories. We emphasize that achieving gains on Cityscapes dataset is challenging because this is a heavily researched benchmark. The previous academic competition elicited gains of $0.8\%$ in mIOU from [97] to [6] over the span of one year.

| Method | road | sidewalk | building | wall | fence | pole | light | sign | vege. | terrain | sky | person | rider | car | truck | bus | train | mbike | bicycle | mIOU |
|---|---|---|---|---|---|---|---|---|---|---|---|---|---|---|---|---|---|---|---|---|
| PSPNet [97] | **98.7** | 86.9 | 93.5 | 58.4 | 63.7 | 67.7 | 76.1 | 80.5 | 93.6 | 72.2 | 95.3 | 86.8 | 71.9 | 96.2 | 77.7 | 91.5 | 83.6 | 70.8 | 77.5 | 81.2 |
| Mapillary Research [6] | 98.4 | 85.0 | 93.7 | **61.8** | **63.9** | 67.7 | 77.4 | 80.8 | 93.7 | 71.9 | 95.6 | 86.7 | 72.8 | 95.7 | 79.9 | 93.1 | **89.7** | 72.6 | 78.2 | 82.0 |
| DeepLabv3+ [14] | **98.7** | 87.0 | **93.9** | 59.5 | 63.7 | **71.4** | **78.2** | **82.2** | **94.0** | 73.0 | **95.9** | 88.0 | 73.3 | 96.4 | 78.0 | 90.9 | 83.9 | 73.8 | 78.9 | 82.1 |
| DPC | **98.7** | **87.1** | 93.8 | 57.7 | 63.5 | 71.0 | 78.0 | 82.1 | **94.0** | **73.3** | 95.4 | **88.2** | **74.5** | **96.5** | **81.2** | **93.3** | 89.0 | **74.1** | **79.0** | **82.7** |

<p align="center">Table 2: Cityscapes <em>test</em> set performance across leading competitive models.</p>

| Method | head | torso | u-arms | l-arms | u-legs | l-legs | bkg | mIOU |
|---|---|---|---|---|---|---|---|---|
| Liang *et al.* [47] | 82.89 | 67.15 | 51.42 | 48.72 | 51.72 | 45.91 | 97.18 | 63.57 |
| Xia *et al.* [89] | 85.50 | 67.87 | 54.72 | 54.30 | 48.25 | 44.76 | 95.32 | 64.39 |
| Fang *et al.* [25] | 87.15 | 72.28 | 57.07 | 56.21 | 52.43 | 50.36 | **97.72** | 67.60 |
| DPC | **88.81** | **74.54** | **63.85** | **63.73** | **57.24** | **54.55** | 96.66 | **71.34** |

<p align="center">Table 3: PASCAL-Person-Part <em>validation</em> set performance.</p>

| Method | aero | bike | bird | boat | bottle | bus | car | cat | chair | cow | table | dog | horse | mbike | person | plant | sheep | sofa | train | tv | mIOU |
|---|---|---|---|---|---|---|---|---|---|---|---|---|---|---|---|---|---|---|---|---|---|
| EncNet [95] | 95.3 | 76.9 | 94.2 | 80.2 | 85.3 | 96.5 | 90.8 | 96.3 | 47.9 | 93.9 | **80.0** | 92.4 | 96.6 | 90.5 | 91.5 | 70.9 | 93.6 | 66.5 | 87.7 | 80.8 | 85.9 |
| DFN [93] | 96.4 | 78.6 | 95.5 | 79.1 | 86.4 | 97.1 | 91.4 | 95.0 | 47.7 | 92.9 | 77.2 | 91.0 | 96.7 | 92.2 | 91.7 | 76.5 | 93.1 | 64.4 | 88.3 | 81.2 | 86.2 |
| DeepLabv3+ [14] | 97.0 | 77.1 | **97.1** | 79.3 | 89.3 | 97.4 | 93.2 | 96.6 | **56.9** | 95.0 | 79.2 | 93.1 | 97.0 | **94.0** | 92.8 | 71.3 | 92.9 | 72.4 | 91.0 | 84.9 | 87.8 |
| ExFuse [96] | 96.8 | **80.3** | 97.0 | **82.5** | 87.8 | 96.3 | 92.6 | 96.4 | 53.3 | 94.3 | 78.4 | 94.1 | 94.9 | 91.6 | 92.3 | **81.7** | **94.8** | 70.3 | 90.1 | 83.8 | 87.9 |
| MSCI [48] | 96.8 | 76.8 | 97.0 | 80.6 | **89.3** | 97.4 | **93.8** | **97.1** | 56.7 | 94.3 | 78.3 | 93.5 | 97.1 | **94.0** | **92.8** | 72.3 | 92.6 | **73.6** | 90.8 | **85.4** | **88.0** |
| DPC | **97.4** | 77.5 | 96.6 | 79.4 | 87.2 | **97.6** | 90.1 | 96.6 | 56.8 | **97.0** | 77.0 | **94.3** | **97.5** | 93.2 | 92.5 | 78.9 | 94.3 | 70.1 | **91.4** | 84.0 | 87.9 |

<p align="center">Table 4: PASCAL VOC 2012 <em>test</em> set performance.</p>

## 4.4 Performance on person part segmentation

PASCAL-Person-Part dataset [16] contains large variation in object scale and human pose annotating six person part classes as well as the background class. We train a model on this dataset employing the same DPC identified during architecture search using the modified Xception network backbone.

Fig. 2 in the supplementary material shows a qualitative visualization of these results and Tab. 3 quantifies the model performance. The DPC architecture achieves state-of-the-art performance of 71.34%, representing a 3.74% improvement over the best state-of-the-art model [25], consistently outperforming other models w.r.t. all categories except the background class. Additionally, note that the DPC model does not require extra MPII training data [1], as required in [89, 25].

## 4.5 Performance on semantic image segmentation

The PASCAL VOC 2012 benchmark [24] (augmented by [32]) involves segmenting 20 foreground object classes and one background class. We train a model on this dataset employing the same DPC identified during architecture search using the modified Xception network backbone.

Fig. 3 in the supplementary material provides a qualitative visualization of the results and Tab. 4 quantifies the model performance on the test set. The DPC architecture outperforms previous state-of-the-art models [95, 93] by more than 1.7%, and is comparable to concurrent works [14, 96, 48]. Across semantic categories, DPC achieves state-of-the-art performance in 6 categories of the 20 categories.

## 5 Conclusion

This work demonstrates how architecture search techniques may be employed for problems beyond image classification – in particular, problems of dense image prediction where multi-scale processing is critical for achieving state-of-the-art performance. The application of architecture search to dense image prediction was achieved through (1) the construction of a recursive search space leveraging innovations in the dense prediction literature and (2) the construction of a fast proxy predictive of the large-scale task. The resulting learned architecture surpasses human-invented architectures across three dense image prediction tasks: scene parsing [18], person-part segmentation [16] and semantic segmentation [24]. In the first task, the resulting architecture achieved performance gains comparable to the gains witnessed in last year's academic competition [18]. In addition, the resulting architecture is more efficient than state-of-the-art systems, requiring half of the parameters and 38% of the computational demand when using deeper Xception [17, 67, 14] as network backbone.

Several opportunities exist for improving the quality of these results. Previous work identified the design of a large and flexible search space as a critical element for achieving strong results [101, 52,

100, 65]. Expanding the search space further by increasing the number of branches $\mathcal{B}$ in the dense prediction cell may yield further gains. Preliminary experiments with $\mathcal{B} > 5$ on the scene parsing data suggest some opportunity, although random search in an exponentially growing space becomes more challenging. The use of intelligent search algorithms such as reinforcement learning [3, 99], sequential model-based optimization [61, 52] and evolutionary methods [81, 69, 59, 90, 53, 68] may be leveraged to further improve search efficiency particularly as the space grows in size. We hope that these ideas may be ported into other domains such as depth prediction [80] and object detection [70, 55] to achieve similar gains over human-invented designs.

**Acknowledgments**   We thank Kevin Murphy for many ideas and inspiration; Quoc Le, Bo Chen, Maxim Neumann and Andrew Howard for support and discussion; Hui Hui for helping release the models; members of the Google Brain, Mobile Vision and Vizier team for infrastructure, support and discussion.

## Footnotes

[1]An implementation of the proposed model will be made available at `https://github.com/tensorflow/models/tree/master/research/deeplab`.

[2]The small search space consists of all possible combinations of the 5 parallel branches of the ASPP architecture – a top ranked architecture for dense prediction [12]. There exist $2^5 - 1 = 31$ potential arrangements of these parallel pathways.

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
