[Supplementary Material]

# Searching for Efficient Multi-Scale Architectures for Dense Image Prediction
# – Supplementary Material –

**Liang-Chieh Chen**     **Maxwell D. Collins**     **Yukun Zhu**     **George Papandreou**
**Barret Zoph**     **Florian Schroff**     **Hartwig Adam**     **Jonathon Shlens**
Google Inc.

In this supplementary material, we provide more dataset details, and visualization results of our proposed model on each dataset.

## 1 Details of dense prediction datasets

### 1.1 Cityscapes

The Cityscapes [2] contains high quality pixel-level annotations of 5000 images with size $1024 \times 2048$ (2975, 500, and 1525 for the training, validation, and test sets respectively) and about 20000 coarsely annotated training images. Following the evaluation protocol [2], 19 semantic labels are used for evaluation without considering the void label.

### 1.2 PASCAL-Person-Part

The PASCAL-Person-Part dataset [1] provides detailed part annotations for every person, including Head, Torso, Upper/Lower Arms and Upper/Lower Legs, resulting in six person part classes and one background class. There are 1716 images for training and 1817 images for validation.

### 1.3 PASCAL VOC 2012 segmentation

The PASCAL VOC 2012 benchmark [3] involves segmenting 20 foreground object classes and one background class. The original dataset contains 1464 (*train*), 1449 (*val*), and 1456 (*test*) pixel-level labeled images for training, validation, and testing, respectively. The dataset is augmented by the extra annotations provided by [4], resulting in 10582 (*trainaug*) training images.

# 2 Visualization of model predictions

## 2.1 Citysacpes

Figure 1: Visualization of predictions on the Cityscapes *validation* set.

## 2.2 PASCAL-Person-Part

Figure 2: Visualization of predictions on PASCAL-Person-Part *validation* set.

## 2.3 PASCAL VOC 2012

Figure 3: Visualization of predictions on PASCAL VOC 2012 *validation* set.