[Reviews · NeurIPS 2018]

Reviewer 1



This paper proposes a meta-learning technique for dense image predictions. Going beyond image classification, authors propose to consider problems where multi-scale processing is important. The problem is challenging and important. The proposal is well introduced and motivated. An extensive review of the related work is presented. The paper is clear and easy to follow. The method is well presented and original. The experimental results properly validates the improvement achieved with the proposed techniques in three different tasks. In my opinion, the paper can be accepted as it is.

Reviewer 2



In this paper, authors explore the construction of meta-learning algorithms, through defining a viable search space and corresponding proxy tasks that are specifically designed and assessed for the task of dense prediction/semantic segmentation. The proposed method and the resulting network is tested on three different relevant datasets (Cityscapes, PASCAL VOC 12 and PASCAL person part). Main strengths: + The method improves the state-of-the-art on three datasets including the competitive cityscapes and PASCAL VOC 2012. + Most of the method ingredients (e.g. the proxy tasks) are well-thought-out and explored. + The paper is fairly well-written and is easy to follow. Major weaknesses: - Even though the meta-learning procedure is strived to become as cheap as possible, it is still impossible to be tried by the research groups with small to moderate available computational resources: "We deploy the resulting proxy task with our proposed architecture search space on Cityscapes to explore 28K DPC architectures across 370 GPUs over one week." - The selected space search algorithm (random search) might not to be the most optimal one. Further improvements are possible by incorporating search algorithms that can achieve good solutions more efficiently (e.g. by better incorporating prior knowledge about the search process) Other (minor) remarks: - It would have been useful to get more insights on the objective function landscape. For instance, the distribution of fitness of the explored architectures could have given information about the amount of variations in the objective function, or a diagram of best objective value based on number of architectures explored, could have indicated how far the search needed to continue. - As mentioned in sections 4.4 and 4.5, the same best found DPC on the cityscapes dataset is used for the PASCAL VOC and PASCAL person part datasets and so no dataset-specific meta-learning is performed. However, it is interesting to assess how well the (close-to-)optimal structure in one dataset can generalize to be a (close-to-)optimal structure in the other ones. This assessment becomes relatively important given the intractability of meta-learning for new dataset for majority of researchers.

Reviewer 3



This paper introduced a method to search multi-scale architectures for dense prediction. The authors defined a search space with 81 elementary operations. As training network for dense prediction is slow, they introduced a proxy task that is fast to evaluate and predict the performance in a large-scale training setting. The new architecture is evaluated on three dense prediction tasks: scene parsing, person-part segmentation, and semantic image segmentation. I have several concerns about this paper: -- The authors did not analyze the correlation between training architecture with 30k iterations and training architecture with 90k iterations. This result is important to confirm the approach. -- As the architecture search is based on a random search, the authors should report the standard deviation in their results. In Figure 3 and 4, the variance of the top 50 architectures seems important. I think it could be interesting to give the results every 2k architectures to show the improvement. -- The authors did not cite Neural Fabrics [101] and GridNet [102] that learned architecture for dense prediction. -- The authors showed that it is possible to find good architectures for dense prediction but they did not analyze or give insights why the learned architecture is better than hand-crafted architecture. What is missing in the existing architectures? What I like: + The definition of the proxy task to speed-up the architecture search + The good results on three dense prediction tasks with a more computationally efficient architecture that requires half the parameters and half the computational cost. Other comments: - The final output of the DPC bloc is the concatenation of the of all branch outputs. Did you explore others strategies? How did you choose the number of filters? - What is the speed-up to cache the feature? - The authors should indicate the batch size. [101] Saxena S., Verbeek J. Convolutional Neural Fabrics. In Advances in Neural Information Processing Systems, 2016. [102] Fourure D., Emonet R., Fromont E., Muselet D., Tremeau A., Wolf C. Residual Conv-Deconv Grid Network for Semantic Segmentation. In BMVC, 2017. I have read the author response and I am globally satisfied. I think an analysis to explain what is wrong in hand crafted architectures could significantly improve the quality and the impact of this paper.